# Receipt of healthcare provider advice to quit tobacco use among Indian men

**Prakash Babu Kodali** [1,2*], **Maya Vijayaraghavan** [3]

**1** Department of Public Health and Community Medicine, Central University of Kerala, Kasaragod, Kerala, India, **2** Center for Tobacco Control Research and Education, University of California, San Francisco, California, United States of America, **3** Division of General Internal Medicine, Department of Medicine, University of California, San Francisco, California, United States of America

\* prakashkodali@cukerala.ac.in

## Abstract

### Introduction

Healthcare provider (HCP) advice to quit smoking is an essential component of smoking cessation interventions. The National Family Health Survey (NFHS) is a periodic health survey of the Indian population. In this cross-sectional analysis we compared NFHS 2015−16 to NFHS 2019−21, and examined the prevalence of the HCP advice to quit tobacco and its associated factors among male cigarette and bidi users.

### Methods

The analytic sample included those who had a HCP visit in the past 12 months. We defined quit advice as receiving any advice to quit smoking from a HCP in the last 12 months ("HCP advice to quit"). We compared sociodemographic characteristics of participants between NFHS-4 and 5. We pooled both samples to conduct multivariate logistic regression analysis for the binary outcome of HCP advice to quit, and examined sociodemographic (age, education, caste, occupation), lifestyle and environmental (smoking frequency, alcohol consumption, second hand smoke exposure), health (non-communicable disease diagnosis), and healthcare use (oral examination, health insurance coverage) factors associated with this outcome.

### Results

Among cigarette and bidi users visiting a HCP, advice to quit significantly improved from NFHS-4 (54.2%) to NFHS-5 (56.7%) ($p < 0.025$). Being 45−54 years of age (AOR = 1.67, 95% CI = 1.47-1.89), having a non-communicable disease diagnosis (AOR = 1.80, 95% CI = 1.53-2.12), and smoking >10 cigarettes/bidi per day (AOR = 1.44, 95% CI = 1.29-1.60) were significantly associated with HCP advice to

**Data availability statement:** This study used secondary data of NFHS obtained from the Demographic and Health Surveys program. The authors are not authorized to share the micro-data. Other researchers can access the NFHS survey datasets from the DHS program (https://dhsprogram.com/) through a formal request in the DHS data portal. The authors received no special privileges in accessing the data that other researchers would not have.

**Funding:** The author(s) received no specific funding for this work.

**Competing interests:** The authors have declared that no competing interests exist.

quit. Individuals from Scheduled Tribes were less likely to receive HCP advice to quit (AOR = 0.60, 95% CI = 0.50-0.71).

## Conclusions

While HCP advice to quit increased from NFHS-4 to NFHS-5, it was limited to those who fit the risk factor profile of a middle-aged male with non-communicable diseases. Although the study is limited by self-reported cross sectional data and survey-item differences between NFHS rounds, our findings present a pressing need for interventions to increase HCP advice to quit to other demographics of men who smoke.

---

## Introduction

Combustible tobacco use is associated with chronic non-communicable diseases (NCD) including chronic respiratory diseases, cardiovascular disease, cancer, as well as mental health conditions [1,2]. Individuals smoking combustible tobacco products like cigarettes have a 64% greater risk of all-cause mortality compared to those who never smoked [3]. With an estimated 100 million individuals using combustible tobacco products, India is the second largest contributor to the global population of tobacco users [4,5]. The 2016−17 Global Adult Tobacco Survey (GATS) in India reported that 91.3% of people who smoke believe smoking causes serious illness, and 74.7% believed smoking causes heart disease [6]. While the intention to quit smoking was 55.3%, intention did not translate to successful quitting, as evidenced by the quit ratio (i.e., successful quitting among those who ever smoked) of only 14.2% [7,8]. Research based on nationally representative surveys in 16 major countries in the global tobacco use landscape, including countries like the United States, United Kingdom, India, China, Russia, Bangladesh, reported that India had the lowest successful quit rates among people who have ever smoked combustible tobacco products [9].

Combustible tobacco use in India is primarily characterized by bidi (handmade mini-cigarettes made of cured tobacco flakes wrapped in a dried tendu leaf) and cigarette smoking. Individuals who smoke are primarily male (91%), in working age groups (i.e., 15–64 years), with less than secondary education (64.1%), rural residents (73.3%), and from low-income households [7,10]. Rural residents, individuals belonging to Scheduled Castes/Tribes, which are recognized as historically marginalized groups in Indian society [11], individuals from low-income households, and individuals with low education levels experience limited to no access to healthcare [12]. Tobacco use worsens existing health inequities, making it essential to quit smoking [13]. However, majority of attempts to quit smoking are unaided, [7] and unassisted attempts are known to have a high rate of relapse [14].

Using cessation treatment that addresses physiological and behavioral aspects of nicotine addiction can significantly improve chances of quitting [15]. Individuals who use guideline-recommended cessation support including behavioral counselling, nicotine replacement therapy (NRT) and other forms of pharmacotherapy have more

than double the odds of quitting tobacco use [15]. In the United States and the United Kingdom, promoting smoking cessation treatment through behavioral counselling and pharmacotherapy was also found to be cost-effective [16]. However, in low and middle-income countries, the financial costs associated with providing cessation treatment remain a significant barrier to promoting widespread access to treatment [15].

India's National Tobacco Control Program has been ongoing for the last two decades [15,17] and experiences significant gaps in reaching the population of people who use tobacco and in providing treatment to those users. The 2015−16 National Mental Health Survey of India showed that only 8.2% of tobacco users received effective support to stop tobacco use [18]. The 2016−17 GATS showed that 38.5% of combustible tobacco users attempted to quit smoking [19], but only 13.2% of those quit attempts were supported by cessation services like behavioral counselling, NRT, or other forms of pharmacotherapy [20,21]. In 2023−24, 1.35 million tobacco users received behavioral counselling through the quitline under the National Tobacco Control Program, accounting for less than 1% of the people who smoke in India [22]. One of the primary reasons for the limited use of cessation services is low awareness of treatment options among tobacco users [23,24]. Fragmented referral systems, low risk perception, and lack of user demand also contribute to reduced access to cessation support [25]. The National Tobacco Control Program's initial work was largely focused on health education and anti-tobacco advocacy, and tobacco cessation services through institutionalized approaches like tobacco cessation centers and quitlines were not widespread until recently, primarily due to a limited health systems capacity [17,25].

Given these substantial gaps in health systems delivery of tobacco treatment, there is a role for guideline-recommended counseling that includes brief advice to quit [26,27]. Brief advice to quit defined as a "*physician or healthcare provider advice to stop tobacco use during a routine consultation*" can increase delivery of a cessation intervention during a healthcare visit [15,27]. The WHO report on the Global Tobacco Epidemic, 2019 advocates that "*integrating brief advice into existing primary healthcare systems is one of the first actions countries can take to develop tobacco cessation support*" [15]. From 2009 to 2022, access to cessation services (i.e., NRT, bupropion, varenicline, and quitline) improved in India, though their use in cessation attempts remained unchanged [17,21]. This low use could be due to limited efforts made towards integrating cessation services into healthcare systems.

Studies globally report that the receipt of quit advice is usually higher among the tobacco users who were diagnosed with a chronic NCDs and those with better access to healthcare through health insurance coverage. [28] Individual lifestyle attributes such as tobacco use characteristics, co-use of alcohol were positively associated with receipt of quit advice [29]. Socio-demographic characteristics including age, ethnicity, education, and income are also associated with receipt of quit advice [28,30–32] (Fig 1).

While, studies have reported that tobacco users advised by a healthcare provider (HCP) had more quit attempts, [26] better cessation services use, [33] and higher quit rates, [8,34,35] there are only few studies that have explored receipt of HCP advice to quit among tobacco users in India [29,31,36]. Most tobacco users are male and between the ages of 15 and 54 years; [6] however, these prior studies did not focus on this demographic [29,31,36]. They did not compare changes in the receipt of HCP advice to quit over time, nor did they account for health status (NCD diagnosis), health behaviors (alcohol & smokeless tobacco use), and healthcare use characteristics (health insurance coverage, and oral health examination), that were previously found to be associated with receiving HCP advice to quit in other studies of global tobacco use [28,37,38].

In this study, we explored the Indian National Family Health Surveys (NFHS) (i.e., NFHS-4 [2015−16], and NFHS-5 [2019−21]), the representative demographic surveys of the Indian population to assess the prevalence of receipt of HCP advice to quit and its associated factors among male combustible tobacco users. We hypothesized that receipt of HCP advice to quit would improve between NFHS-4 [2015−16] and NFHS-5 [2019−21] and NCD diagnosis, health behaviors (i.e., frequency of smoking, use of alcohol, smokeless tobacco) and healthcare use (i.e., undergoing oral health examination, and health insurance coverage) would be associated with the receipt of HCP advice to quit.

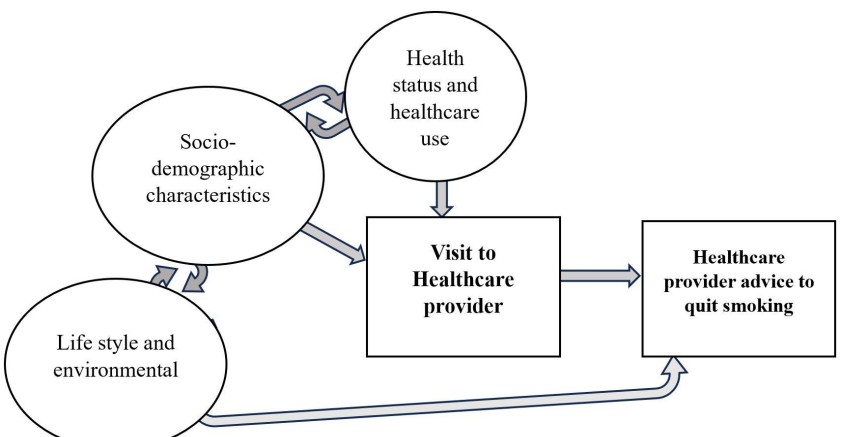

**Fig 1. Conceptual framework explaining receipt of HCP advice to quit among individuals who smoke. Socio-demographic characteristics:** Age, Sex, Place of residence, Education, Occupation, Caste, Wealth Index. **Life style and environmental factors:** Frequency of smoking, Smokeless tobacco use, Alcohol use, Exposure to second hand smoke. **Health status & health care use:** Self-reported NCD diagnosis, History of undergoing an oral examination, Health insurance coverage.

## Materials and methods

### Study design

We conducted a cross-sectional analysis of waves four (2015−16) and five (2019−21) of the Indian NFHS. The Indian NFHS is part of the Demographic and Health surveys (DHS) [39], administered at five-year intervals. The NHFS provides information on health status, healthcare needs, and health behaviors including tobacco use.

The NFHS uses standardized protocols for sample selection, data collection, and data management, allowing for national, state, and district-level estimates [39,40]. The NFHS surveys used in this analysis employed a uniform multi-stage stratified sampling design. The districts (administrative units within each Indian state) were stratified into rural and urban strata. In rural strata, the villages (lowest administrative regions in rural areas) were selected as primary sampling units (PSU), while census enumeration blocks were selected as PSUs in urban regions. The PSUs which represented a cluster were sampled using probability proportionate to size sampling approach. Further, from each PSU, a representative sample of households were randomly sampled to be included in district, state, and dried blood sample modules. All men in the age groups of 15−54 years residing in the households sampled for the state module were surveyed. A total of 112,122 (sampled from 109,318 households), and 101,839 (sampled from 91,400 households) men were surveyed in NFHS-4 & 5 with response rates of 92% and 88% respectively [39,40].

The NFHS collected data using structured survey schedules administered by trained field investigators employing the Computer Assisted Personal Interviewing approach. The surveys were administered in regional languages. The details about survey instruments, data collection, data management and quality control are provided in the published reports of NFHS-4 & 5 [39,40].

### Study sample

We conducted this study among a subsample of adult males who smoked and visited a HCP in the last 12 months. Participants self-reported using cigarettes or bidis either daily or occasionally in the past one year at the time of survey. We included only cigarettes and bidi users, as the prevalence of other combustible forms of tobacco was low (<2%)

and they overlapped with cigarette and bidi use [4]. Participants who responded affirmatively to currently smoking cigarettes or smoking bidis every day or some days were categorized as current combustible tobacco users, and those who reported not using these products at the time of the survey were non-users. Of the 29,761 combustible tobacco users from NFHS-4, 5,080 visited a HCP in the past year, and were included in the analysis. From the 21,708 combustible tobacco users from NFHS-5, 3729 who visited a HCP in the past year were included in the analysis. The analytic sample comprised of a pooled sample of 8809 current combustible tobacco users who visited a HCP in the 12 months prior to NFHS surveys (Fig 2).

### Study variables

**HCP advice to quit.** Participants responded to the following questions i) *"In the last 12 months, have you visited a doctor or other healthcare provider?"*, and ii) *"During any of these visits, were you advised to quit smoking or using tobacco in any other form?".* Individuals who answered "Yes" to both survey items were considered to have received HCP advice to quit (S1 File).

**Covariates.** We grouped covariates into i) socio-demographic characteristics, ii) lifestyle and environmental factors, and iii) health status and healthcare use.

**Sociodemographic characteristics.** Participants self-reported information on age, which we categorized into three groups: 15–29 years, 30–44 years and 45–54 years. Participants self-reported whether they were from urban or a rural place of residence. Participants self-reported their highest education attained with response options including no education, primary, secondary, or higher education. Participants self-reported the caste they belonged to including i) scheduled castes (SC), ii) scheduled tribes (ST), iii) other backward castes (OBC), and iv) other castes including those that do not fall into either of SC, ST, or OBC [41]. Participants reported their occupation which we categorized as i) not employed; ii) agriculture, unskilled/skilled manual jobs, and domestic jobs and iii) professional/technical/managerial,

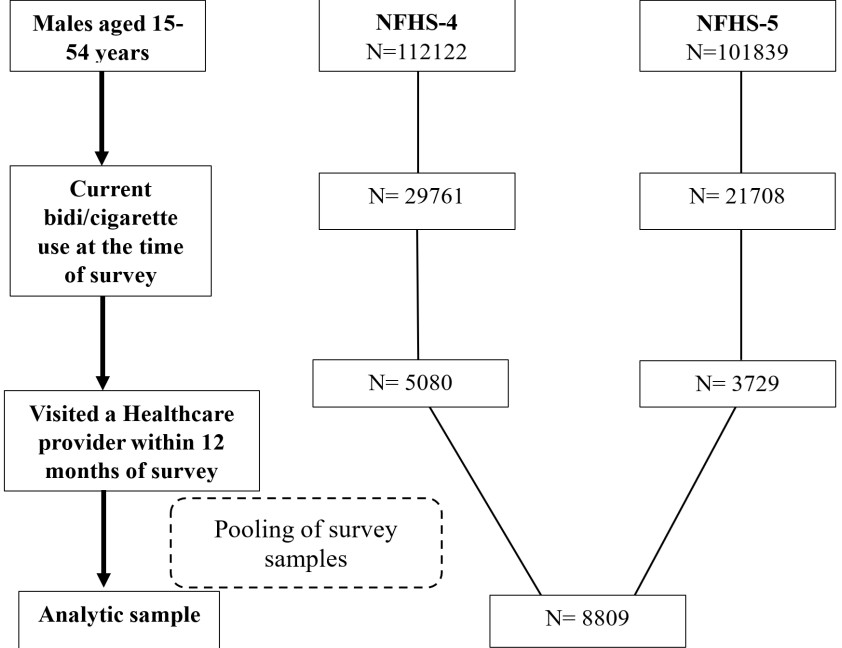

**Fig 2. Flowchart outlining the selection of analytic sample from the NFHS survey samples.** NFHS = National Family Health Survey; N = Sample Size.

clerical, and sales sector [41]. We categorized the wealth index into quintiles (i) poorest, ii) poorer, iii) middle, iv) richer, and v) richest) using previously defined NFHS survey methodology [39,40].

**Lifestyle and environmental factors.**  Based on the items "*On average, how many cigarettes do you currently smoke each day?",* and "*On average, how many bidis do you currently smoke each day?*", the frequency of smoking cigarettes/bidi was categorized as '≤ 10 cigarettes/bidi per day' and '>10 cigarettes/bidi per day' [42]. We categorized participants who self-reported use of smokeless tobacco products (Khaini or Paan) daily or less than daily at the time of the survey as smokeless tobacco users. Participants who self-reported alcohol consumption (Tadi madi, country liquor, beer, wine, hard liquor or others) at least once a month were categorized as alcohol users. Participants were considered exposed to secondhand smoke if they reported that someone other than themselves smoked in their presence in the last 30 days [43].

**Health status and healthcare use.**  Participants self-reported whether they had a previous diagnosis of NCD including diabetes, chronic respiratory diseases, heart disease, and cancer. They reported whether they had any oral examination in their lifetime (NHFS-4) or had been screened for oral cancer (NHFS-5). These items were pooled to develop a single binary item 'ever had an oral examination'. Participants reported whether they were uninsured or had public or private health insurance coverage.

## Data analysis

We report unweighted frequencies & weighted percentages to describe the study sample characteristics for NFHS-4 and NFHS-5. We report the absolute percent change in outcome and independent variables between the two surveys and used chi-square statistic to test differences between the two waves [44]. We pooled the study samples from NHFS-4 and NHFS-5 for bivariate and multivariate logistic regression analysis and reported adjusted odds ratios and 95% confidence intervals for the binary outcome of HCP advice to quit. We explored sociodemographic, lifestyle, environmental, health status, healthcare use factors as well as survey year associated with HCP advice to quit among those who visited a HCP in the past year. We conducted all analyses using Statistical Package for Social Sciences (SPSS) version 27 (IBM Corp. Released 2020. IBM SPSS Statistics for Windows, Version 27.0. Armonk, NY: IBM Corp)

## Ethics Statement

This study involved the analysis of anonymized secondary data from the NFHS-4 & 5 surveys freely accessible from the DHS website (https://dhsprogram.com/) accessed on 30th October 2024. The original NFHS surveys were approved by the Institutional Review Board of International Institute of Population Sciences, Mumbai and the ICF. The NFHS surveys obtained informed consent from the participants. Given the study involved secondary analysis of de-identified microdata, separate ethical approval was not warranted. Consent to participate was not applicable due to the secondary nature of the work, and the researchers do not have access to any personally identifiable information.

## Results

The study sample comprised of rural residents (NFHS-4: 66.8%; NFHS-5: 69.1%), who worked in agriculture, unskilled/skilled manual, or domestic jobs (NFHS-4: 73.8%; NFHS-5:71.8%) and had an education of up to secondary level (NFHS-4:88.4%; NFHS-5:86.3%, Table 1).

### Receipt of HCP advice to quit among combustible tobacco users

The proportion who received HCP advice to quit was 54.2% (95%CI = 52.1%−56.4%) in NFHS-4 and 56.7% (95%CI = 53.6%−59.7%) in NFHS-5, a 2.5% increase that was statistically significant ($\chi^2$ = 4.99, df = 1, p = 0.025) (Fig 3).

**Table 1. Characteristics of the combustible tobacco users visiting healthcare providers in NFHS 4 & 5 surveys.**

| Sample characteristics | Current combustible tobacco users visiting HCP | | | Absolute change from NFHS-4–5 | |
| --- | --- | --- | --- | --- | --- |
| | Pooled Sample (n = 8809) % | NFHS-4 (n = 5080) % | NFHS-5 (n = 3729) % | Percentage change | p-value# |
| **Age Group** | | | | | |
| 15-29 years | 29.1 | 29.1 | 29.2 | 0.1 | 1.000 |
| 30-44 years | 41.5 | 41.5 | 41.5 | 0.0 | |
| 45-54 years | 29.3 | 29.3 | 29.3 | 0.0 | |
| **Place of residence** | | | | | |
| Rural | 67.9 | 66.8 | 69.1 | 2.3 | 0.024 |
| Urban | 32.1 | 33.2 | 30.9 | −2.3 | |
| **Education level** | | | | | |
| No education | 19.5 | 20.4 | 18.5 | −1.9 | 0.003 |
| Primary | 18.8 | 19.4 | 18 | −1.4 | |
| Secondary | 49.2 | 48.7 | 49.9 | 1.2 | |
| Higher education | 12.5 | 11.6 | 13.7 | 2.1 | |
| **Caste** | | | | | |
| Scheduled Castes | 23.8 | 23.5 | 24 | 0.5 | <0.001 |
| Scheduled Tribes | 9.1 | 9.1 | 9 | −0.1 | |
| Other Backward Castes | 35.8 | 37.5 | 33.7 | −3.8 | |
| Other caste groups* | 31.4 | 29.8 | 33.3 | 3.5 | |
| **Occupational group** | | | | | |
| Unemployed | 7.8 | 9.4 | 5.9 | −3.5 | 0.001 |
| Agriculture, unskilled/skilled manual jobs, and domestic jobs | 50.7 | 73.8 | 71.8 | −2.0 | |
| Professional/technical/managerial, clerical, and sales sector | 41.5 | 16.8 | 22.3 | 5.5 | |
| **Wealth Index** | | | | | |
| Poorest | 20.5 | 18.6 | 22.8 | 4.2 | <0.001 |
| Poorer | 22.7 | 22.7 | 22.7 | 0.0 | |
| Middle | 21.8 | 22.4 | 21.1 | −1.3 | |
| Richer | 19.6 | 19.6 | 19.6 | 0.0 | |
| Richest | 15.3 | 16.6 | 13.8 | −2.8 | |
| **Frequency of smoking** | | | | | |
| ≤ 10 cigarettes/bidi per day | 52.4 | 50.5 | 54.3 | 3.8 | 0.025 |
| > 10 cigarettes/bidi per day | 63.7 | 62.5 | 65.9 | 3.4 | |
| **Uses smokeless tobacco** | | | | | |
| Yes | 25.6 | 27.2 | 23.7 | −3.5 | <0.001 |
| **Alcohol consumption** | | | | | |
| Yes | 51.6 | 52.7 | 50.2 | −2.5 | 0.024 |
| **Secondhand smoke exposure** | | | | | |
| Yes | 87.5 | 88.5 | 86.2 | −2.3 | 0.001 |
| **Diagnosed with NCD$** | | | | | |
| Yes | 11.9 | 8.8 | 15.7 | 6.9 | <0.001 |
| **Ever had oral examination** | | | | | |
| Yes | 7.8 | 13.1 | 1.4 | −11.7 | <0.001 |

*(Continued)*

**Table 1.** (Continued)

| Sample characteristics | Current combustible tobacco users visiting HCP | | | Absolute change from NFHS-4–5 | |
| --- | --- | --- | --- | --- | --- |
| | Pooled Sample (n = 8809) % | NFHS-4 (n = 5080) % | NFHS-5 (n = 3729) % | Percentage change | p-value# |
| **Covered by health insurance** | | | | | |
| Yes | 34.5 | 30.9 | 39 | 8.1 | <0.001 |

NFHS = National Family Health Survey; n = unweighted frequencies; HCP = healthcare provider; NCD = Non communicable disease; % = Weighted percentages; Caste refers to the system of social hierarchy in India. Other caste groups* are those that do not fall under any of the castes of scheduled castes (SC), scheduled tribes (ST) or other backward castes (OBC). $ NCDs included four non-communicable diseases (Diabetes, Asthma, Heart Disease, Cancer). #The comparison of sample characteristics between NFHS-4 & 5 was conducted using Chi-Square tests. p-values less 0.05 was considered statistically significant.

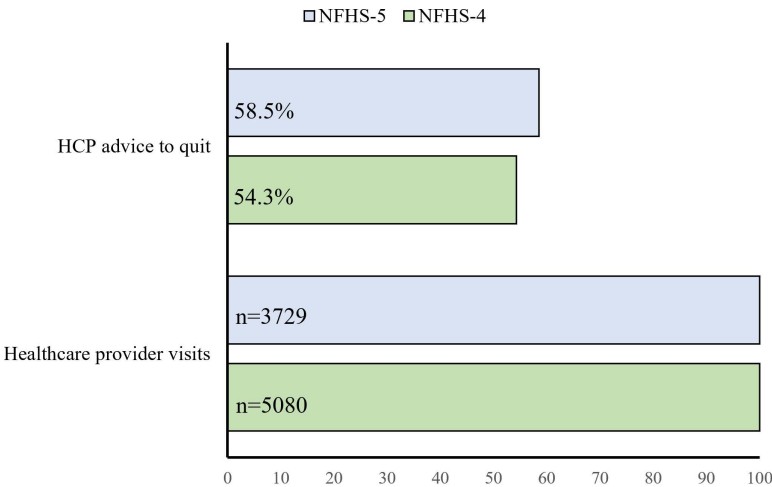

**Fig 3. Receipt of HCP advice to quit among combustible tobacco users: changes from NFHS-4 to NFHS-5.** NFHS: National Family Health Survey; HCP = Healthcare providers; Combustible tobacco users were defined as individuals who reported to be daily or less than daily users of either cigarettes or bidis. **Note:** The 56.7% of combustible tobacco users receiving HCP advice to quit in NFHS-5 account for 9.7% of all male combustible tobacco users surveyed during NFHS-5 (including those who visited HCP, and not visited HCP). The 54.2% of combustible tobacco users receiving HCP advice to quit in NFHS-4 account for 9.3% of all male combustible tobacco users surveyed in NFHS-4 (including those who visited HCP and not visited HCP). The estimates reported are weighted percentages.

## Factors associated with receipt of HCP advice to quit

In multivariable analysis, participants in the NHFS-5 were more likely to have received HCP advice to quit than those in NHFS-4 (Adjusted odds ratio [AOR] 1.27, 95% CI 1.14-1.42). Participants in the 45−54 years age group (AOR 1.67, 95% CI 1.47-1.89), with a diagnosis of a NCD (AOR 1.80, 95% 1.53-2.12), those smoking >10 cigarettes/bidi daily compared to ≤ 10 cigarettes/bidi per day (AOR 1.44, 95% CI 1.29-1.60), and who had ever received an oral examination (AOR 1.57, 95% CI 1.32-1.87) were more like to have received HCP advice to quit compared to their counterparts (Table 2). Those who belong to scheduled tribes (AOR 0.60, 95% CI 0.50-0.71) and other backward castes (AOR 0.80, 95% CI 0.72-0.90) were less likely to have received HCP advice to quit compared to their counterparts.

**Table 2. Factors associated with receiving HCP advice to quit among the pooled sample of current combustible tobacco users visiting healthcare providers during NFHS-4 & NFHS-5 (n = 8809).**

| Independent variables | | HCP advice to quit | | AOR (95% CI) | p-value |
|---|---|---|---|---|---|
| | | Received (n = 4512) | Not-received (n = 4297) | | |
| **Survey round** | n | % | % | | |
| NFHS-4 (ref) | 5080 | 54.20 | 45.80 | | |
| NFHS-5 | 3729 | 56.70 | 43.30 | 1.27 (1.14-1.42) | <0.001 |
| **Age Group** | | | | | |
| 15-29 years (ref) | 2458 | 45.70 | 54.30 | | |
| 30-44 years | 3956 | 56.80 | 43.20 | 1.39 (1.25-1.55) | <0.001 |
| 45-54 years | 2395 | 62.90 | 37.10 | 1.67 (1.47-1.89) | <0.001 |
| **Place of residence** | | | | | |
| Urban (ref) | 2370 | 56.60 | 43.40 | | |
| Rural | 6439 | 54.80 | 45.20 | 0.90 (0.81-1.01) | 0.072 |
| **Education level** | | | | | |
| No education (ref) | 1575 | 60.40 | 39.60 | | |
| Primary | 1561 | 54.70 | 45.30 | 0.84 (0.73-0.97) | 0.016 |
| Secondary | 4703 | 54.80 | 45.20 | 0.89 (0.75-0.97) | 0.062 |
| Higher | 970 | 50.30 | 49.70 | 0.72 (0.60-0.87) | 0.001 |
| **Caste** | | | | | |
| Other caste groups (ref) | 2323 | 58.00 | 42.00 | | |
| Scheduled Castes | 1723 | 58.40 | 41.60 | 1.02 (0.90-1.16) | 0.742 |
| Scheduled Tribes | 1913 | 45.00 | 55.00 | 0.60 (0.50-0.71) | <0.001 |
| Other Backward Castes | 2850 | 53.60 | 46.40 | 0.80 (0.72-0.90) | <0.001 |
| **Occupational group** | | | | | |
| Unemployed (ref) | 751 | 45.20 | 54.80 | | |
| Agriculture, unskilled/skilled manual jobs, and domestic jobs | 3560 | 55.80 | 44.20 | 1.23 (1.03-1.48) | 0.023 |
| Professional/technical/managerial, clerical, and sales sector | 4498 | 56.60 | 43.40 | 1.39 (1.17-1.65) | <0.001 |
| **Wealth Index** | | | | | |
| Richest (ref) | 1287 | 55.80 | 44.20 | | |
| Poorest | 1721 | 55.10 | 44.90 | 1.00 (0.83-1.20) | 0.976 |
| Poorer | 2198 | 52.60 | 47.40 | 0.86 (0.72-1.03) | 0.103 |
| Middle | 1971 | 56.70 | 43.30 | 1.03 (0.87-1.22) | 0.7059 |
| Richer | 1632 | 56.80 | 43.20 | 1.05 (0.90-1.23) | 0.535 |
| **Frequency of smoking** | | | | | |
| ≤ 10 cigarettes/bidi per day (ref) | 6435 | 52.40 | 47.60 | | |
| > 10 cigarettes/bidi per day | 2374 | 63.70 | 36.30 | 1.44 (1.29-1.60) | <0.001 |
| **Uses smokeless tobacco^** | | | | | |
| No(ref) | 6338 | 55.90 | 44.10 | | |
| Yes | 2471 | 53.80 | 46.20 | 01.02 (0.92-1.14) | 0.658 |
| **Consumes Alcohol$** | | | | | |
| No (ref) | 4156 | 55.80 | 44.20 | | |
| Yes | 4653 | 54.90 | 45.10 | 1.02 (0.92-1.14) | 0.741 |
| **Secondhand smoke exposure** | | | | | |
| No(ref) | 937 | 52.90 | 47.10 | | |
| Yes | 7872 | 55.70 | 44.30 | 1.10 (0.96-1.25) | 0.175 |
| **Diagnosed with an NCD*** | | | | | |
| No (ref) | 8087 | 53.70 | 46.30 | | |

*(Continued)*

**Table 2.** (Continued)

| Independent variables | | HCP advice to quit | | AOR (95% CI) | p-value |
| --- | --- | --- | --- | --- | --- |
| | | Received (n = 4512) | Not-received (n = 4297) | | |
| Yes | 722 | 71.20 | 28.80 | 1.80 (1.53-2.12) | <0.001 |
| **Ever had oral examination†** | | | | | |
| No (ref) | 7943 | 54.60 | 45.40 | | |
| Yes | 866 | 64.40 | 35.60 | 1.57 (1.32-1.87) | <0.001 |
| **Covered by health insurance** | | | | | |
| No (ref) | 5887 | 54.80 | 45.20 | | |
| Yes | 2992 | 56.40 | 43.60 | 1.05 (0.95-1.15) | 0.344 |

Dependent variable: HCP advice to quit: Received, not advised to quit (ref); ref = reference category; n = unweighted frequencies, % = weighted percentages, AOR = Adjusted Odds Ratios, CI = Confidence Interval; Other caste groups are those that do not fall under any of the castes of scheduled castes (SC), scheduled tribes (ST) or other backward castes (OBC). Caste refers to the system of social hierarchy in India. ^Smokeless tobacco analyzed included Khaini & Paan with Tobacco, not using smokeless tobacco is the reference category; *NCDs included four non-communicable diseases (Diabetes, Asthma, Heart Disease, Cancer), not reporting an NCD is the reference category; †Oral examination was assessed in NFHS-4 using the item "*Ever had oral cavity examination*"; while it was assessed in NFHS-5 using the item "*Ever undergone an oral cavity examination for oral cancer*", not having an oral examination is the reference category. Statistical significance was decided at a p-value of ≤0.05.

## Discussion

In this study of a nationally representative sample of male adults who smoked and who visited a HCP in the past year, there was 2.5% absolute increase in the proportion who received HCP advice to quit in the five years between the NHFS-4 and NHFS-5 surveys. Only half of the male tobacco users who visited HCP received advice to quit. While the increase between two waves equates to an additional million individuals, this this still leaves eight million tobacco users who visited a HCP, but did not receive advice to quit. A similar trend was observed in the GATS survey, which recorded an annual increase of less than 1% between GATS-1 (2009−10) and GATS-2 (2016−17) for HCP advice to quit [36]. While statistically significant, this rate of increase remains insufficient given the large number of tobacco users in India [20,31]. Further, state wise variations in HCP advice to quit exist with states like Tamil Nadu, Kerala, Telangana, and Andhra Pradesh reporting a higher proportion of HCP advice to quit compared to states like Odisha, West Bengal, Assam and Manipur (S2 File). Tobacco use patterns differ across Indian states, and healthcare providers engagement in screening for tobacco use has been documented to be lower in high prevalence states [Odisha, West Bengal, Assam, Manipur etc] compared to low prevalence states [Kerala, Tamil Nadu, Telangana, Andhra Pradesh] [6,45]. Healthcare visits offer an opportunity to provide cessation support, and our findings highlight an important avenue for interventions to train and increase capacity among HCP to screen for tobacco use and provide brief advice to quit during routine health encounters [25].

A previous study showed that combustible tobacco users in India have a low rate of use of cessation services in quit attempts (i.e., 9% for counselling, and 4.4% using pharmacotherapy) [20]. A recent outpatient department survey in Maharashtra, India reported that 79.2% of patients were not asked about their tobacco use status nor advised to quit [46], highlighting that awareness of asking about tobacco use and providing advice to quit is low among HCPs [47]. Further, HCPs may be practicing in healthcare ecosystems where cessation support systems may not be available [21], and cessation medications may be unaffordable [48], which may influence their clinical practice behavior [49]. The 2022 report on Tobacco Control in India stresses the importance of (a) strengthening the demand for cessation support by providing education to tobacco users on the availability of cessation treatment, (b) ensuring the availability of cessation support through the routine healthcare delivery systems, and (c) establishing two way referral between HCPs, and cessation support systems like quit lines and tobacco cessation centers [17].

Consistent with previous studies, our study showed that HCPs were more likely to advise older patients with NCDs [36,50], highlighting a need for a universal screening and tobacco treatment approach in healthcare. Younger patients, in particular, may benefit from receiving HCP advice to quit, as quitting smoking before the age of 35 reverses the harmful effects of smoking equivalent to that of a never user [51]. Universal screening would also address the gap in screening and HCP advise to quit for the 40% of patients who smoke but are not being advised to quit.

Individuals from scheduled tribes, other backward castes, and those reporting to be unemployed are traditionally known to have poor access to healthcare [52,53], and have limited access to quit advice even after reaching the HCP [36]. Healthcare needs of these groups are met primarily by public sector providers or local rural medical practitioners [54,55], who may face time constraints around discussing patients' tobacco use behaviors and offering advice to quit, or are not trained to do so [56,57].

Ever undergoing an oral examination including that for oral cancer screening is associated with being advised to quit tobacco use. Despite differences in the framing of questions between NFHS-4 and NFHS-5 [58], the positive association between undergoing an oral cavity examination and receiving advice to quit highlights the potential role of accessible oral healthcare services in strengthening tobacco control. The National Oral Health Programme (2014) envisages to improve access to oral healthcare services, while the operational guidelines for Health and Wellness Centers (HWCs) integrates tobacco cessation to oral healthcare services [59,60]. Although these policy initiatives and service integrations are promising, the utilization of oral healthcare services among Indian adults remains low at approximately 24.0%, with disparities across southern (30.0%), central (18.9%), western (17.6%) and northern regions (10.6%) [61].

A significant improvement in the receipt of healthcare provider (HCP) advice to quit between the two NFHS rounds may also reflect broader initiatives aimed at expanding healthcare access during 2015–2021. During this period, health insurance coverage in some Indian states nearly doubled [39], and the Government of India upgraded the existing primary healthcare facilities to include HWCs with tobacco cessation services as part of NCD care [62,63]. The number of tobacco cessation centers grew from ~19 in 2002 to >2000 in 2021 [25], and India launched the National Tobacco Quitline Services to provide quitline services in all major Indian languages [25]. Tobacco control also became a key area of focus as part of the larger COVID-19 response [64]. However, further studies are needed to investigate the effective translation of tobacco control responses in improving HCP advice to quit tobacco use and cessation service delivery.

## Limitations

This study has several limitations. First, the analysis could not distinguish between different types of healthcare providers (e.g., physicians, nurses, dentists, or traditional medicine practitioners) due to the lack of provider-specific information in the NFHS dataset. Second, NFHS data are based on self-reported responses, which may be subject to recall bias and differences in participants understanding of survey questions. Third, variations in survey instruments across NFHS rounds specifically, the assessment of oral cavity examination in NFHS-4 and oral cancer screening in NFHS-5 may have resulted in measurement bias, as these variables were combined into a single variable for oral examination. Also due to non-availability of specific variables in the data set we were unable to account for the number of healthcare visits, primary reason for healthcare visit, type of healthcare facilities, or provider characteristics all of which could have impacted on HCP advice. The NFHS-5 (2019−21) was conducted during the period of COVID-19, which could have impacted health seeking, oral examinations, and patient provider interactions. Future studies are needed to assess how these factors influence HCPs advice to quit tobacco use. Additionally, the cross-sectional study design limits our ability to draw causal inferences from the observed associations.

## Conclusions

Despite these limitations, our findings extend the literature by showing that about half of people who smoke and who visit a HCP do not receive advice to quit. Findings have implications for universal screening for tobacco use and increasing

provider training and capacity to offer advice to quit at all healthcare encounters, particularly those providers working in public healthcare systems. Given that tobacco use in India is concentrated among socio-economically marginalized populations, focused investments and education are needed to especially empower individuals from low socioeconomic groups to seek cessation support and provide accessible cessation services delivered through public healthcare system.

## Supporting information

**S1 File. Variable Transformation Sheet.**
(DOCX)

**S2 File. State level differences in the receipt of HCP advice to quit among combustible tobacco users.**
(DOCX)

## Acknowledgments

The authors gratefully acknowledge the International Institute of Population Sciences and the DHS Program for providing access to the National Family Health Survey data. PBK was a Visiting Scholar at the University of California San Francisco, Center for Tobacco Control Research and Education and received support from the United States-India Educational Foundation through the Fulbright-Nehru Postdoctoral Fellowship.

## Author contributions

**Conceptualization:** Prakash Babu Kodali, Maya Vijayaraghavan.

**Data curation:** Prakash Babu Kodali.

**Formal analysis:** Prakash Babu Kodali.

**Methodology:** Prakash Babu Kodali, Maya Vijayaraghavan.

**Supervision:** Maya Vijayaraghavan.

**Visualization:** Prakash Babu Kodali.

**Writing – original draft:** Prakash Babu Kodali.

**Writing – review & editing:** Prakash Babu Kodali, Maya Vijayaraghavan.

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
