## [Decision Letter · Decision Letter 0]

30 Dec 2025

PONE-D-25-29325Receipt of healthcare provider advice to quit tobacco use among Indian menPLOS One

Dear Dr. Kodali,

Thank you for submitting your manuscript to PLOS ONE. After careful consideration, we feel that it has merit but does not fully meet PLOS ONE’s publication criteria as it currently stands. Therefore, we invite you to submit a revised version of the manuscript that addresses the points raised during the review process.

We look forward to receiving your revised manuscript.

Kind regards,

George Kuryan

Academic Editor

PLOS One

**Journal Requirements:**

2. For studies involving third-party data, we encourage authors to share any data specific to their analyses that they can legally distribute. PLOS recognizes, however, that authors may be using third-party data they do not have the rights to share. When third-party data cannot be publicly shared, authors must provide all information necessary for interested researchers to apply to gain access to the data. (https://journals.plos.org/plosone/s/data-availability#loc-acceptable-data-access-restrictions)

Reviewers' comments:

Reviewer's Responses to Questions

**Comments to the Author**

1. Is the manuscript technically sound, and do the data support the conclusions?

Reviewer #1: Yes

2. Has the statistical analysis been performed appropriately and rigorously? 

Reviewer #1: Yes

3. Have the authors made all data underlying the findings in their manuscript fully available?

Reviewer #1: Yes

4. Is the manuscript presented in an intelligible fashion and written in standard English?

Reviewer #1: Yes

5. Review Comments to the Author

Reviewer #1: This manuscript addresses an important and understudied aspect of tobacco control in India—the receipt of healthcare provider (HCP) advice to quit among male smokers, using two large NFHS datasets (4 & 5). The topic aligns well with national and global tobacco control priorities, leveraging high-quality, nationally representative data.

The paper is clearly written, the results are relevant, and the public-health implications are important. However, there are methodological limitations, conceptual gaps, and missing clarifications that require attention.

Introduction

1. The NFHS questions regarding the HCP visit and advice

The definition does not specify which type of HCP (doctor, nurse, AYUSH, dentist, etc.). The study assumes HCPs are comparable. In the Indian context, it is a major oversimplification. Also the study could be biased as the information on the HCPs are based on the information from the patient.

It is ideal to categorise the HCPs and might give an input on the variations in the smokers in each category

2. Under the health status subheading, two questions are mentioned

“ever had oral cavity examination”, “ever undergone oral cancer screening”

These are not equivalent constructs, yet are pooled as one variable for oral cancer screening in the table. This could be seen as a methodological flaw. It can be mentioned as a limitation, or by analysing after excluding this variable

Methods

3. NFHS 4 mentions 4 NCDs, and NFHS 5 mentions 6 NCDs; pooling these NCDs can falsely increase the association. Were any corrections made on the same?

Results:

4. Figure 1 shows many determinants for the advice to quit. However potential confounders are missed in analysis such as: Frequency of smoking or cigarettes/bidis per day (Heavier smokers are more likely to mention it and to be advised to quit).; Number of healthcare visits (More visits → higher probability of receiving advice); Type of facility (public vs private); Quit advice frequency differs widely across sectors in India. Provider characteristics (gender, training, cadre), although there should not be a difference, some people tend to listen to a male HCP more.

5. Needs clarity on tables and results, need a bit organizing

Minor Issues

1. Abstract is strong but could briefly mention limitations.

2. Improve grammar & discussion to reduce repetition.

6. PLOS authors have the option to publish the peer review history of their article (what does this mean?). If published, this will include your full peer review and any attached files.

Reviewer #1: **Yes:** Thomas Antony

---

## [Author Response · Author response to Decision Letter 1]

5 Feb 2026

Reviewer #1: This manuscript addresses an important and understudied aspect of tobacco control in India—the receipt of healthcare provider (HCP) advice to quit among male smokers, using two large NFHS datasets (4 & 5). The topic aligns well with national and global tobacco control priorities, leveraging high-quality, nationally representative data.

The paper is clearly written, the results are relevant, and the public-health implications are important. However, there are methodological limitations, conceptual gaps, and missing clarifications that require attention.

Author response: We appreciate the reviewer’s comment. We revised the manuscript accordingly. Where suggested revisions were not feasible due to data limitations, we explicitly acknowledged these constraints and reported them in the limitations section.

Introduction

1. The NFHS questions regarding the HCP visit and advice

The definition does not specify which type of HCP (doctor, nurse, AYUSH, dentist, etc.). The study assumes HCPs are comparable. In the Indian context, it is a major oversimplification. Also the study could be biased as the information on the HCPs are based on the information from the patient.

It is ideal to categorise the HCPs and might give an input on the variations in the smokers in each category

Author response: We agree that disaggregating Healthcare provider (HCP) by type (e.g., doctors, nurses, AYUSH practitioners, dentists) and reporting advice to quit tobacco by provider category would add important nuance, particularly in the Indian context, where care-seeking is heterogeneous.

However, we are unable to undertake such an analysis due to inherent limitations of the National Family Health Survey (NFHS) data. Specifically, advice to quit tobacco use is captured only through the following linked items: (i) “In the last 12 months, have you visited a doctor or other healthcare provider?” and (ii) “During any of these visits, were you advised to quit smoking or using tobacco in any other form.” These items do not specify the type of HCP associated with the advice to quit.

While NFHS includes separate questions on interactions with specific provider types (such as doctors, nurses, or AYUSH providers), these items pertain to different recall periods (e.g., last 3 months) or distinct service domains (e.g., family planning or immunization). As a result, these variables could not be harmonized with the tobacco cessation advice variable in a methodologically valid manner. We have recognized this as a major limitation of our study in line 350-352.

2. Under the health status subheading, two questions are mentioned

“ever had oral cavity examination”, “ever undergone oral cancer screening”

These are not equivalent constructs, yet are pooled as one variable for oral cancer screening in the table. This could be seen as a methodological flaw. It can be mentioned as a limitation, or by analysing after excluding this variable

Author response: We agree with the reviewer that “ever had an oral cavity examination” and “ever undergone oral cancer screening” are not strictly equivalent constructs. As the reviewer correctly notes, NFHS-4 captured oral examination using a broader item, whereas NFHS-5 introduced a more specific question on oral cancer screening.

To maintain comparability across survey rounds and avoid excluding a potentially relevant oral health–related variable, we retained this measure and operationalized it conservatively as “ever had an oral examination.” We acknowledge that this variable likely captures a heterogeneous set of encounters, ranging from general oral examinations to formal oral cancer screening, and therefore should not be interpreted as a precise measure of oral cancer screening uptake.

We considered excluding this variable; however, doing so would have omitted information on prior oral health contact, which may plausibly influence healthcare provider–initiated tobacco cessation advice, particularly in a high-tobacco-burden setting such as India. To minimize misinterpretation, we have avoided referring to this variable as “oral cancer screening” and have explicitly acknowledged the non-equivalence of constructs and potential misclassification as a study limitation (Lines 354–357).

Methods

3. NFHS 4 mentions 4 NCDs, and NFHS 5 mentions 6 NCDs; pooling these NCDs can falsely increase the association. Were any corrections made on the same?

Author response: To ensure comparability, we restricted the NCD variable to the four conditions common to both survey rounds (diabetes, asthma, heart disease, and cancer). We revised our analysis by applying the updated variable for ‘Diagnosed with an NCD’ and made corresponding changes to our results section [line 276-282] and table 2 in the revision.

Results:

4. Figure 1 shows many determinants for the advice to quit. However potential confounders are missed in analysis such as: Frequency of smoking or cigarettes/bidis per day (Heavier smokers are more likely to mention it and to be advised to quit).; Number of healthcare visits (More visits → higher probability of receiving advice); Type of facility (public vs private); Quit advice frequency differs widely across sectors in India. Provider characteristics (gender, training, cadre), although there should not be a difference, some people tend to listen to a male HCP more.

Author response: We developed our conceptual framework (figure 1), by reviewing published studies, NFHS survey questionnaires and respective data sets. Based on your suggestion, we included the item ‘Frequency of smoking cigarettes/bidi per day’ by developing it from independent items capturing ‘number of cigarettes smoked per day’ and ‘number of bidis smoked per day’. We captured it as a binary item with responses ‘≤ 10 cigarettes/bidi per day’ and ‘>10 cigarettes/bidi’ per day (Cigarette Smoking Reduction and Health Risks: A Systematic Review and Meta-analysis - PubMed). We revised our Figure 1, Table 1, Table 2, Results and discussion by including this important variable.

With respect to the items such as number of healthcare visits, type of facility, and provider characteristics, we are unable to include them since the NFHS surveys did not capture these items. We have acknowledged this in the limitations, and indicated the need for future studies by including these items in lines 357-359.

5. Needs clarity on tables and results, need a bit organizing

Author response: We revised and better organized the tables and Results section in line with the suggested corrections to improve clarity. Specifically, in Table 2, we now present unweighted frequencies for all independent variables alongside weighted percentages for the outcome variable (i.e., receipt of healthcare provider advice to quit), improving clarity and interpretability.

Minor Issues

1. Abstract is strong but could briefly mention limitations.

Author response: Thank you we have reported the study limitations in the abstract of the manuscript (line 53 & 54).

2. Improve grammar & discussion to reduce repetition.

Author response: We improved grammar and reduced repetition in the discussion.

---

## [Decision Letter · Decision Letter 1]

27 Feb 2026

PONE-D-25-29325R1Receipt of healthcare provider advice to quit tobacco use among Indian menPLOS One

Dear Dr. Kodali, Thank you for submitting your manuscript to PLOS ONE. After careful consideration, we feel that it has merit but does not fully meet PLOS ONE’s publication criteria as it currently stands. Therefore, we invite you to submit a revised version of the manuscript that addresses the points raised during the review process.

Please submit your revised manuscript by Apr 13 2026 11:59PM. If you will need more time than this to complete your revisions, please reply to this message or contact the journal office at plosone@plos.org. Please include the following items when submitting your revised manuscript:

We look forward to receiving your revised manuscript.

Kind regards,

George Kuryan

Academic Editor

PLOS One

Journal Requirements:

Reviewers' comments:

Reviewer's Responses to Questions

**Comments to the Author**

1. If the authors have adequately addressed your comments raised in a previous round of review and you feel that this manuscript is now acceptable for publication, you may indicate that here to bypass the “Comments to the Author” section, enter your conflict of interest statement in the “Confidential to Editor” section, and submit your "Accept" recommendation.

Reviewer #2: (No Response)

Reviewer #3: All comments have been addressed

2. Is the manuscript technically sound, and do the data support the conclusions?

Reviewer #2: Partly

Reviewer #3: Yes

3. Has the statistical analysis been performed appropriately and rigorously? 

Reviewer #2: Yes

Reviewer #3: Yes

4. Have the authors made all data underlying the findings in their manuscript fully available?

Reviewer #2: Yes

Reviewer #3: Yes

5. Is the manuscript presented in an intelligible fashion and written in standard English?

Reviewer #2: Yes

Reviewer #3: Yes

6. Review Comments to the Author

Reviewer #2: The current study explores the important topic of tobacco-cessation counseling by way

of HCP advice. The use of large-scale representative data, the statistical analysis and

the results are acceptable. However, there are gaps in discussion that need to be addressed to

make clear recommendations. The points below are noted for the consideration of the

authors.

1. Ref: Rayhan Rahman, Illias Sheikh, Yash Merchant, Ankita Mathur, Toufiq Noor, Vini Mehta,

Oral cancer screening in India: a secondary analysis of disparities using national family health survey data, Oral Oncology, Volume 169, 2025, 107632, ISSN 1368-8375, https://doi.org/10.1016/j.oraloncology.2025.107632.

(https://www.sciencedirect.com/science/article/pii/S1368837525004610)

The above study uses NFHS data and explores an important point overlooked in the current study- the prevalence of tobacco usage- smoke/smokeless differs widely between states as observed in the NFHS survey too.

It mentions: “Screening participation is directly related to lower tobacco prevalence in the population. The study found that people from high-tobacco-prevalence states were 42 % less likely to be screened (Adjusted Odds Ratio (AOR): 0.58, p < 0.01) than those from low-prevalence states”.

This finding has also been noted in the GATS study.

This wide variation between the Southern Indian states and Union territories as compared to Northern Indian states can be mentioned and even included in the analysis.

2.Ref:

https://www.researchgate.net/publication/368273529_Utilization_of_dental_care_services_among_adult_Indian_population_A_meta-analysis_of_evidence_from_2011-2022

The above is a meta-analysis that estimates dental care seeking among Indian adults and discusses various factors including the North – South divide in oral care seeking, findings worth noting and mentioning in the discussion.

3. The National Oral Health program launched by the MOHFW in 2014 and its scope could be included in the discussion.

4. An important finding from NFHS 4 vs NFHS 5- as noted in a recent study (referenced below) is that “The proportion of men reporting oral cavity examinations was 9% (out of 112,122 interviewed men) in NFHS-4, which reduced to 0.2% (out of 716,443 interviewed men) in NFHS-5”

Ref: Learning from the Indian National Family Health Survey to assess population based oral, cervix and breast cancer screening in low-and-middle income countries. Borkotoky, Kakoli et al.

The Lancet Regional Health - Southeast Asia, Volume 30, 100483

5. Ref: Ramanadhan S, Xuan Z, Choi J, Mahtani SL, Minsky S, Gupte H, Mandal G, Jagiasi D, Viswanath K. Associations between sociodemographic factors and receiving "ask and advise" services from healthcare providers in India: analysis of the national GATS-2 dataset. BMC Public Health. 2022 Nov 18;22(1):2115. doi: 10.1186/s12889-022-14538-2. PMID: 36401241; PMCID: PMC9673333.

The findings of the above study, analyzing the GATS-2 dataset, are relevant and would do well to be included in the discussion for the current study

6. The fact that the timing of the NFHS 5 survey was during the pandemic when rampant COVID screening and testing were underway could have impacted health seeking and also contributed to the noticed increase in self-reports of oral examinations during NFHS 5 may be worth mentioning.

Reviewer #3: Its a nice piece of work, done on a very pertinent and buring topic of the contemprory times. However, certain aspects needs clarificatrion.

1. The study describes that only 50% of tobaco users attending the health care facility were given advice to quit. In this context it is pertinent to note the primery reason for attending the health care facility. Since it will have a large bearing on giving advice. For instance in case of accidents or emergencies it is unlikely that HCP will provide advice to quit. They will be focused more on stabilizing the patients. Same is the case with attendence for resons which are unrelated to tobacco use, emaple - Surgical conditions etc.

2. Findings will be more reliable if data is provided to demonstarte that the visit to health facility was due to a conditon/ health state which arised due to tobacco use. since such information is not availabe in present study, policy impications becomes unjustifiable.

7. PLOS authors have the option to publish the peer review history of their article (what does this mean?). If published, this will include your full peer review and any attached files.

Reviewer #2: No

Reviewer #3: No

---

## [Author Response · Author response to Decision Letter 2]

20 Mar 2026

Reviewer #2: The current study explores the important topic of tobacco-cessation counseling by way

of HCP advice. The use of large-scale representative data, the statistical analysis and

the results are acceptable. However, there are gaps in discussion that need to be addressed to

make clear recommendations. The points below are noted for the consideration of the

authors.

Author response: Dear Reviewer, thank you for your careful review of our manuscript and the suggestions you have made. We agree with the revisions suggested and had made the corrections in the manuscript. We ensured to include the suggested articles in our discussion and strengthen our findings and their interpretation by including a state-level differences in receipt of HCP advice to quit as a supplementary file. We look forward to your positive consideration of our manuscript.

1. Ref: Rayhan Rahman, Illias Sheikh, Yash Merchant, Ankita Mathur, Toufiq Noor, Vini Mehta,

Oral cancer screening in India: a secondary analysis of disparities using national family health survey data, Oral Oncology, Volume 169, 2025, 107632, ISSN 1368-8375, https://doi.org/10.1016/j.oraloncology.2025.107632.

(https://www.sciencedirect.com/science/article/pii/S1368837525004610)

The above study uses NFHS data and explores an important point overlooked in the current study- the prevalence of tobacco usage- smoke/smokeless differs widely between states as observed in the NFHS survey too.

It mentions: “Screening participation is directly related to lower tobacco prevalence in the population. The study found that people from high-tobacco-prevalence states were 42 % less likely to be screened (Adjusted Odds Ratio (AOR): 0.58, p < 0.01) than those from low-prevalence states”.

This finding has also been noted in the GATS study.

This wide variation between the Southern Indian states and Union territories as compared to Northern Indian states can be mentioned and even included in the analysis.

Author response:

Thank you for the suggestion. We have included the state-wise variations in HCP advise to quit as a supplementary file. We have discussed the variations in HCP advice to quit in connection with the observations made by Rahman et al (2025), and GATS India survey 2016-17. The changes were made to lines 306 to 312 and supplementary file 2 was added.

2.Ref:

https://www.researchgate.net/publication/368273529_Utilization_of_dental_care_services_among_adult_Indian_population_A_meta-analysis_of_evidence_from_2011-2022

The above is a meta-analysis that estimates dental care seeking among Indian adults and discusses various factors including the North – South divide in oral care seeking, findings worth noting and mentioning in the discussion.

Author response: Thank you for the suggestion. We have revised our discussion section to include the suggested study in the discussion section in lines 350-353.

3. The National Oral Health program launched by the MOHFW in 2014 and its scope could be included in the discussion.

Author response: Thank you for the suggestion. We have incorporated a discussion on the scope of the National Oral Health Programme in the revised manuscript (Lines 348–353).

4. An important finding from NFHS 4 vs NFHS 5- as noted in a recent study (referenced below) is that “The proportion of men reporting oral cavity examinations was 9% (out of 112,122 interviewed men) in NFHS-4, which reduced to 0.2% (out of 716,443 interviewed men) in NFHS-5”

Ref: Learning from the Indian National Family Health Survey to assess population based oral, cervix and breast cancer screening in low-and-middle income countries. Borkotoky, Kakoli et al.

The Lancet Regional Health - Southeast Asia, Volume 30, 100483

Author response: Thank you for the suggestion, we have included the suggested study in our discussion in lines 345-346

5. Ref: Ramanadhan S, Xuan Z, Choi J, Mahtani SL, Minsky S, Gupte H, Mandal G, Jagiasi D, Viswanath K. Associations between sociodemographic factors and receiving "ask and advise" services from healthcare providers in India: analysis of the national GATS-2 dataset. BMC Public Health. 2022 Nov 18;22(1):2115. doi: 10.1186/s12889-022-14538-2. PMID: 36401241; PMCID: PMC9673333.

The findings of the above study, analyzing the GATS-2 dataset, are relevant and would do well to be included in the discussion for the current study

Author response: Thank you for the suggestion, we have included the suggested study in our discussion.

6. The fact that the timing of the NFHS 5 survey was during the pandemic when rampant COVID screening and testing were underway could have impacted health seeking and also contributed to the noticed increase in self-reports of oral examinations during NFHS 5 may be worth mentioning.

Author response: Thank you for this important observation. We agree that the timing of NFHS-5 during the COVID-19 pandemic may have influenced health-seeking behaviour, oral examinations, and interactions with healthcare providers. We have acknowledged this as a limitation of the study in the revised manuscript (Lines 378–379).

Reviewer #3: Its a nice piece of work, done on a very pertinent and buring topic of the contemprory times. However, certain aspects needs clarificatrion.

1. The study describes that only 50% of tobaco users attending the health care facility were given advice to quit. In this context it is pertinent to note the primery reason for attending the health care facility. Since it will have a large bearing on giving advice. For instance in case of accidents or emergencies it is unlikely that HCP will provide advice to quit. They will be focused more on stabilizing the patients. Same is the case with attendence for resons which are unrelated to tobacco use, emaple - Surgical conditions etc.

Author response: Thank you for this valuable observation. We agree that the primary reason for visiting a healthcare facility can significantly influence the likelihood of receiving advice to quit tobacco. However, the NFHS surveys do not capture information on the reason for healthcare visits. We have acknowledged this as a limitation in the revised manuscript (Lines 375–378) and have interpreted our findings with appropriate caution.

2. Findings will be more reliable if data is provided to demonstarte that the visit to health facility was due to a conditon/ health state which arised due to tobacco use. since such information is not availabe in present study, policy impications becomes unjustifiable.

Author response: Thank you for this important comment. We acknowledge that the dataset does not include information on the specific reasons for healthcare visits, and therefore we are unable to ascertain whether these visits were related to tobacco use. This has been explicitly recognized as a limitation in the revised manuscript [lines 375 to 378].

---

## [Decision Letter · Decision Letter 2]

23 Apr 2026

Receipt of healthcare provider advice to quit tobacco use among Indian men

PONE-D-25-29325R2

Dear Dr. Kodali

We’re pleased to inform you that your manuscript has been judged scientifically suitable for publication and will be formally accepted for publication once it meets all outstanding technical requirements.

Kind regards,

George Kuryan

Academic Editor

PLOS One

Additional Editor Comments (optional):

Reviewers' comments:

Reviewer's Responses to Questions

**Comments to the Author**

1. If the authors have adequately addressed your comments raised in a previous round of review and you feel that this manuscript is now acceptable for publication, you may indicate that here to bypass the “Comments to the Author” section, enter your conflict of interest statement in the “Confidential to Editor” section, and submit your "Accept" recommendation.

Reviewer #2: All comments have been addressed

Reviewer #3: All comments have been addressed

2. Is the manuscript technically sound, and do the data support the conclusions?

Reviewer #2: Yes

Reviewer #3: Yes

3. Has the statistical analysis been performed appropriately and rigorously? 

Reviewer #2: I Don't Know

Reviewer #3: Yes

4. Have the authors made all data underlying the findings in their manuscript fully available?

Reviewer #2: Yes

Reviewer #3: Yes

5. Is the manuscript presented in an intelligible fashion and written in standard English?

Reviewer #2: Yes

Reviewer #3: Yes

6. Review Comments to the Author

Reviewer #2: The authors have addressed adequately the comments raised by the reviewer. Considering the complexity of the demography under consideration, this study can provide important insights for studies in the future.

Reviewer #3: Appreciate the authers work. Meticulously done on very neglected aspect .

all comments got adressed

I dont have any further comments to make.

7. PLOS authors have the option to publish the peer review history of their article (what does this mean?). If published, this will include your full peer review and any attached files.

Reviewer #2: No

Reviewer #3: No

---

## [Editor Report · Acceptance letter]

PONE-D-25-29325R2

PLOS One

Dear Dr. Kodali,

I'm pleased to inform you that your manuscript has been deemed suitable for publication in PLOS One. Congratulations! Your manuscript is now being handed over to our production team.

Kind regards,

on behalf of

Professor George Kuryan

Academic Editor

PLOS One